# Intelligent Fault Diagnosis of Industrial Robot Based on Multiclass Mahalanobis-Taguchi System for Imbalanced Data

**DOI:** 10.3390/e24070871

**Published:** 2022-06-24

**Authors:** Yue Sun, Aidong Xu, Kai Wang, Xiufang Zhou, Haifeng Guo, Xiaojia Han

**Affiliations:** 1Key Laboratory of Networked Control Systems, Chinese Academy of Sciences, Shenyang 110169, China; sunyue@sia.cn (Y.S.); wangkai@sia.cn (K.W.); zhouxiufang@sia.cn (X.Z.); 2Shenyang Institute of Automation, Chinese Academy of Sciences, Shenyang 110169, China; 3Institutes for Robotics and Intelligent Manufacturing, Chinese Academy of Sciences, Shenyang 110169, China; 4University of Chinese Academy of Sciences, Beijing 100049, China; 5College of Electrical and Information Engineering, Liaoning Institute of Science and Technology, Benxi 117004, China; guohaifeng@sia.cn; 6Intelligent Robot Research Center of Zhejiang Laboratory, Hangzhou 311100, China; hanxiaojia@zhejianglab.com

**Keywords:** intelligent fault diagnosis, industrial robots, imbalanced data, imbalance ratio, multiclass Mahalanobis-Taguchi system

## Abstract

One of the biggest challenges for the fault diagnosis research of industrial robots is that the normal data is far more than the fault data; that is, the data is imbalanced. The traditional diagnosis approaches of industrial robots are more biased toward the majority categories, which makes the diagnosis accuracy of the minority categories decrease. To solve the imbalanced problem, the traditional algorithm is improved by using cost-sensitive learning, single-class learning and other approaches. However, these algorithms also have a series of problems. For instance, it is difficult to estimate the true misclassification cost, overfitting, and long computation time. Therefore, a fault diagnosis approach for industrial robots, based on the Multiclass Mahalanobis-Taguchi system (MMTS), is proposed in this article. It can be classified the categories by measuring the deviation degree from the sample to the reference space, which is more suitable for classifying imbalanced data. The accuracy, G-mean and F-measure are used to verify the effectiveness of the proposed approach on an industrial robot platform. The experimental results show that the proposed approach’s accuracy, F-measure and G-mean improves by an average of 20.74%, 12.85% and 21.68%, compared with the other five traditional approaches when the imbalance ratio is 9. With the increase in the imbalance ratio, the proposed approach has better stability than the traditional algorithms.

## 1. Introduction

As the core industrial equipment, industrial robots are used by more and more manufacturing enterprises to replace people for high-precision and high-repeatability production work. Due to the long-term operation of industrial robots in harsh working environments and some unforeseen factors, faults occur from time to time. The high maintenance costs and long maintenance cycles bring huge economic losses to enterprises, and the life safety of technicians is even threatened [1]. The fault diagnosis of industrial robots, as the key content of servitization of the equipment manufacturing industry, helps to reduce production loss and ensure the production safety of enterprises.

The industrial robot fault diagnosis approaches [2] are mainly classified into model-based approaches [3], knowledge-based approaches [4], and data-driven approaches. At present, data-driven approaches are increasingly favored by scholars. For the fault diagnosis approaches of machine learning [5,6,7] and deep learning [8,9], the premise of its feasibility is that the training datasets for various modes are balanced. However, the biggest challenge for the fault diagnosis of an industrial robot in real industrial environments is the imbalanced data between categories. In the classification research for imbalanced data, conventional approaches, such as SVM [10] and C4.5 [11], are more biased toward the majority categories and ignore the minority categories in order to maximize the global diagnostic accuracy, resulting in a decrease in the diagnosis accuracy of minority categories. However, the misclassification cost of the minority categories is often greater [12].

### 1.1. Research Status and Existing Problems for Imbalanced Data

To address the bias problem of classification approaches for imbalanced data, scholars have studied many improvement approaches. The data augmentation strategy is an effective means to expand the scale of data to achieve balance. Data resampling [13,14] is the most representative data augmentation approach. The approach of changing the number of samples may introduce noise or remove important information. Generative adversarial network (GAN) [15,16] and Variational autoencoder (VAE) [17] are data generation models that have emerged in recent years. The GAN and VAE use a few samples for training to generate low-quality fault samples and inaccurate diagnostic accuracy without the support of big data. At the same time, a large number of computational resources are required. The diagnostic efficiency is low, which is not friendly to practical applications. While expanding the scale of data, the introduction of cost-sensitive learning [10,18], single-class learning [19,20], and ensemble learning [21,22] can also solve the imbalance problem. Among them, cost-sensitive learning is used to introduce different misclassification costs for different categories. It can effectively deal with the classification problem of imbalanced data with the goal of minimizing the overall misclassification cost. In practical applications, the real sample distribution is uncertain. It is difficult to estimate the true misclassification cost. The effectiveness of cost-sensitive learning cannot be confirmed. The single-class learning approach is only used to train and model the target samples. Then, it identifies this class of samples from the test samples. It does not need to identify the non-target samples, which greatly improves the classification efficiency. However, it is prone to overfitting when the minority category samples are trained as target samples, which leads to a decrease in the generalization ability. The ensemble learning approach improves the classification performance of the imbalanced data classification problems by integrating multiple classifiers. Although it has better classification results, it is complex, and the computation time is long [23].

### 1.2. The Approach Proposed in This Article

In view of the above problems, this article adopted the Mahalanobis-Taguchi system (MTS) for intelligent fault diagnosis research on industrial robots with imbalanced data. MTS is a multivariate pattern recognition approach proposed by Dr Genichi Taguchi, a famous Japanese quality engineer [24]. It is a data-based analysis approach with simple principles and convenient applications. MTS makes its decision attribution through Mahalanobis distance (MD). The Taguchi approach is used to filter out the effective features and optimize the classification problem. It can achieve the reduction dimension of true meaning. The traditional approach of directly determining the category of the sample. However, MTS is a measurement method that determines the category of the sample by constructing the Mahalanobis space of the benchmark category to calculate the deviation degree from the test sample to the Mahalanobis space [25]. MTS builds a multi-dimensional scale based on a single category sample rather than relying on the whole training set; therefore, MTS [26] is more suitable for the classification problem of imbalanced data. In addition, MTS does not need to consider the relevant costs of misclassified samples. It is not easy to over-fit. Furthermore, diagnosis theory is simple and consumes less computational resources.

The construction of a single Mahalanobis space in MTS can only solve the binary classification problem. There is a low recognition accuracy for the multi-classification fault diagnosis problems. However, in the practical application process, there are various fault categories for industrial robots, which are not only limited to binary classification problems. Therefore, this article proposes a fault diagnosis approach for industrial robots based on the MMTS under the condition of imbalanced data. A classification study of the multiple fault modes of industrial robot reducer bearings is carried out.

The rest of this article is organized as follows. An intelligent fault diagnosis approach for industrial robots based on the MMTS is introduced in Section 2. The experimental platform and comparative analysis are shown in Section 3. Finally, the conclusion is drawn in Section 4, followed by references.

## 2. Intelligent Fault Diagnosis Approach for Industrial Robots Based on MMTS

### 2.1. Construction of Multiple Mahalanobis Spaces

Different from the two-class MTS, the MMTS is used to construct the corresponding Mahalanobis space based on each category, respectively. $MS^{\left(t\right)}$ is defined as the *t*-th Mahalanobis space, where t=1,2,…,P, *P* is the number of mode categories. The category samples for constructing the Mahalanobis space are usually regarded as the normal samples. In this article, the normal samples of the t-th category of the industrial robot system are obtained. Its feature parameters are defined as $X_{i}^{\left(t\right)}$, where i=1,2,…,p, p represents the number of feature parameters. $X_{ij}^{\left(t\right)}$ is the observed value of the *i*-th parameter on the *j*-th sample, where j=1,2,…,m, and m represents the number of the normal samples. The mean $\bar{X_{i}^{\left(t\right)}}$ and the standard deviation $S_{i}^{\left(t\right)}$ of each feature parameter for the normal sample of the t-th mode are calculated and normalized. The formula is shown in (1).
(1)$$Z_{ij}^{\left(t\right)}=\frac{X_{ij}^{\left(t\right)}-\bar{X_{i}^{\left(t\right)}}}{S_{i}^{\left(t\right)}}t=1,2,\ldots ,P\;,\;i=1,2,\ldots ,p,j=1,2,\ldots ,m\;$$
where
$$\bar{X_{i}^{\left(t\right)}}=\frac{1}{m}\sum_{j=1}^{m}X_{ij}^{\left(t\right)}$$
$$S_{i}^{\left(t\right)}=\sqrt{\frac{\sum_{j=1}^{m}{\left(X_{ij}^{\left(t\right)}-\bar{X_{i}^{\left(t\right)}}\right)}^{2}}{m-1}}$$

Finally, the MD of the j-th sample is calculated by (2).
(2)$$MD_{j}^{\left(t\right)}=\frac{1}{p}Z_{j}^{\left(t\right)}\cdot corr^{{\left(t\right)}^{-1}}\cdot Z_{j}^{\left(t\right)}{}^{{}^{T}}$$
where $Z_{j}^{\left(t\right)}=[Z_{j1}^{\left(t\right)},Z_{j2}^{\left(t\right)},\ldots ,Z_{jp}^{\left(t\right)}]$, $Z_{j}^{\left(t\right)}{}^{{}^{T}}$ is the transpose of $Z_{j}^{\left(t\right)}$. The correlation coefficient matrix formula is shown in (3).
(3)$$corr^{\left(t\right)}=\frac{1}{m-1}\sum_{j=1}^{m}Z_{j}^{\left(t\right)}{}^{{}^{T}}Z_{j}^{\left(t\right)}$$

The inverse matrix approach used to calculate MD is expressed in the above. Then, when there is a high correlation between the feature parameters, the determinant of its correlation coefficient matrix tends to be 0. The calculation results of the inverse matrix are inaccurate, resulting in an inaccurate calculation of MD. Although the multi-source signal fusion approach can obtain the comprehensive state information of the robot system, it also has some redundant information. Therefore, it is necessary to propose a more effective and stable approach to solve the problem of a strong correlation among the features of the industrial robot system. Currently, some scholars use the Schmidt orthogonalization approach instead of the inverse matrix approach. However, the Schmidt orthogonalization approach needs to consider the order of the feature variables in the orthogonalization. The optimized features will also change if the order is changed. Other scholars use the adjoint matrix approach to solve MD. However, for the larger the better type of signal-to-noise (S/N) ratio, there are defects in the feature optimization stage. The feature parameters cannot be optimized. Therefore, the M–P generalized inverse-matrix approach with better robustness is used to solve the strong correlation problem [27]. The calculation formula of MD based on the M–P generalized inverse matrix is shown in (4).
(4)$$MD_{j}^{\left(t\right)}=\frac{1}{p}Z_{j}^{\left(t\right)}\cdot corr^{{\left(t\right)}^{+}}\cdot Z_{j}^{\left(t\right)}{}^{{}^{T}}$$
where $corr^{{\left(t\right)}^{+}}$ is the M–P generalized inverse-matrix of the correlation coefficient matrix $corr^{\left(t\right)}$.

### 2.2. Validation of Mahalanobis Space

When a certain category is used as the benchmark to construct the Mahalanobis space, the samples of the remaining categories are regarded as abnormal samples. The abnormal samples containing the above feature parameters are selected and normalized by the mean and standard deviation of the benchmark Mahalanobis space. Then, the MD of the abnormal samples is calculated by combining the correlation coefficient matrix. It is known that the MD of normal samples is around 1. If the MD of the abnormal samples is significantly larger than the MD of the normal samples, the constructed Mahalanobis space is effective. Otherwise, it is necessary to re-select the normal samples or feature parameters that can represent the kind of modes to construct an effective Mahalanobis space.

### 2.3. Optimization of Mahalanobis Space Based on Orthogonal Arrays (OAs) and S/N Ratios

For the industrial robot systems, redundant feature parameters may exist in the initial feature set that is constructed. In this section, OAs and S/N ratios will be used to select the useful features for each category of Mahalanobis space by evaluating the gain. The OAs is used to determine the minimum number of trials for feature combinations. It not only saves trial costs, but also guarantees performance. The rows in the OAs represent each feature combination, and the columns represent the features. In this article, an OAs of $L_{d^{u}}(d^{f})$ type is used, where $f=(d^{u}-1)/(d-1)$. d represents the number of levels, d=2. A two-level OAs is defined as containing p-feature parameters, 0<p≤f. The p features are placed in the first p columns of the OAs. The number of levels for each feature corresponding to each trial is 2. In the OAs, “1” means the feature is selected, “2” means the feature is not selected. The feature with the level of 1 is selected for each trial to construct the Mahalanobis space. The $MD_{k}{}^{\left(t\right)}$ of each abnormal sample is calculated according to the n features selected in each trial where k=1,2,…n.

In order to evaluate the robustness of the feature parameters, the S/N ratio is introduced into the orthogonal trial as an evaluation index for screening the feature parameters. In this article, the promising large S/N ratio is selected. The S/N ratio of each trial is calculated as shown in (5).
(5)$$SN^{\left(t\right)}=-10\mathrm{lg}(\frac{1}{n}\sum_{k=1}^{n}\frac{1}{MD_{k}{}^{\left(t\right)}})$$

When all the trails are completed, for the feature parameter $X_{i}^{\left(t\right)}$, $SN_{i}^{+}{}^{\left(t\right)}$ represents the average S/N ratio of the feature selected to participate in the trail. $SN_{i}^{-}{}^{\left(t\right)}$ represents the average S/N ratio of the feature that is not selected to participate in the trail. $\Delta_{i}^{\left(t\right)}$ represents the increment of the S/N ratio. The calculation formula is shown in (6). When $\Delta_{i}^{\left(t\right)}$ is positive, the feature parameter $X_{i}^{\left(t\right)}$ is retained. When $\Delta_{i}^{\left(t\right)}$ is negative, the feature parameter $X_{i}^{\left(t\right)}$ is removed. Thus, the task of feature optimization is completed, which greatly improves the efficiency of the fault diagnosis.
(6)$$\Delta_{i}^{\left(t\right)}=SN_{i}^{+}{}^{\left(t\right)}-\;SN_{i}^{-}{}^{\left(t\right)}$$

### 2.4. Fault Mode Recognition of MMTS

Based on the optimized features in Section 2.3, a new Mahalanobis space, $MS\_new^{\left(t\right)}$, is reconstructed. The test sample is also optimized for features. The $MD_{x}^{\left(t\right)}$ of this test sample to the Mahalanobis spaces are calculated separately. From the multiclass discriminant criterion, it is known that the test sample belongs to the mode category corresponding to the minimum MD identified by (7).
(7)$$\min(MD_{x}^{\left(1\right)},MD_{x}^{\left(2\right)},\ldots MD_{x}^{\left(t\right)},\ldots ,MD_{x}^{\left(P\right)})$$

## 3. Test Platform and Test Results

### 3.1. Industrial Robot Data Acquisition Platform

In this section, the research work of the industrial robot fault diagnosis study is realized with the help of the SR10AL SIASUN industrial robot and the NI data acquisition system. The robot is a six-axis industrial robot with a rated load of 10 KG. It is known that the weak link of this robot is the five-axis harmonic reducer; that is, the five-axis harmonic reducer gradually degrades or even fails after long-term operation. The internal drive-end bearing is the weak link of the reducer. Therefore, the drive-end bearing in the five-axis reducer is taken as the object to study the fault diagnosis of the industrial robot reducer bearing in this section. The research framework is shown in Figure 1.

The acoustic emission signal, current signal and vibration signal are collected by the acoustic emission sensor, current transformer and vibration sensor. The rotation angle information is obtained with the help of the PXI system. The sampling frequencies of vibration signal, acoustic emission signal and current signal are 12 kHz, 20 kHz and 10 kHz, respectively. According to the fault mechanism analysis of the industrial robot reducer bearing, it is known that the vibration signal and acoustic emission signal are the main signal sources to reflect the fault information. The acoustic emission sensor and vibration sensor are installed in the manipulator shell on the side of the five-axis reducer, which are close to the location of the bearing to be tested. Therefore, multiple time-domain and time-frequency domain information are extracted to obtain more comprehensive fault information. The current transformer is installed at the output of the servo driver in the control cabinet to collect the current signal. The current signal is used as an auxiliary diagnostic basis to extract the RMS features that best reflect the current characteristics. The industrial robot platform is shown in Figure 2.

In this section, seven modes (the health and six fault modes) of the bearing are diagnosed. The modes are listed in Table 1. The bearing schematic diagram is shown in Figure 3. The operating conditions of the industrial robot are shown in Table 2. The first column is the symbol of the work condition, the second column is the percentage of the maximum speed, and the third column is the load condition. The half load is 5 kg, which is half of the rated load. The no-load is 0.

### 3.2. Intelligent Fault Diagnosis Test of Industrial Robot Based on MMTS Algorithm

In the actual industrial environment, there is a problem with imbalanced data in the field of industrial robot fault diagnosis. The normal data are far more than the fault data. For this reason, the industrial robot intelligent fault diagnosis test bed for the imbalanced data is needed. The motion range of the five-axis joint of the industrial robot is set from −90° to 90°. The relatively stable signal, in the range of −45° to 45°, is selected for analysis. Every 1024 points are divided into one sample. A total of 900 training samples are obtained, including 540 normal samples and 60 training samples for each of the remaining six fault modes. The imbalance ratio is 9, which is consistent with the imbalanced data problem, where the imbalance ratio refers to the ratio of the sample number of the majority categories to each minority category in the training sample. The test samples for each fault mode are 60. Taking the *D*_70_*BAN* working condition as an example, the sample schematic diagram of the seven fault modes of the reducer bearing is shown in Figure 4. Among them, (a)~(b) are the acoustic emission signal, the current signal and the vibration signal, respectively. The relevant features are extracted, and the initial feature set is constructed. Finally, MMTS is used for feature optimization and fault mode recognition.

#### 3.2.1. The Construction and Effective Verification of Mahalanobis Space

The initial Mahalanobis space is constructed with NO, BF05, BF1, IF05, IF1, OF1, and OF2 as benchmarks, respectively. The remaining categories are used as the abnormal samples for validation. The results are shown in Figure 5. The MD of each benchmark space is around 1. The MD of the remaining abnormal samples is larger or even much larger than the MD of the benchmark space. Therefore, it is proved that the constructed Mahalanobis space is valid.

#### 3.2.2. Optimization of Mahalanobis Space

After the effective Mahalanobis space is constructed, the important feature parameters are selected by OAs and the S/N ratio. In this section, the 12 dimensional feature parameters [F1–F12] = [RMS, K, Vpp, Var, E1, E2, E3, E4, E5, E6, E7, E8] are extracted based on the vibration signal and acoustic emission signal, respectively. The RMS features are extracted based on the current signal. A two-level OAs is constructed based on the normal samples. The 25 dimensional feature parameters are placed in the first 25 columns of the orthogonal array. The average S/N ratio of the feature selected to participate in thetrail, the average S/N ratio of the feature not selected to participate in the trail, and the increment of the S/N ratio are shown in Table 3. Based on the values, the important feature parameters selected are F1, F2, F4, F5, F11, F13, F14, F15, F16, F17, F20, F21, F24. Table 4 lists the subsets selected after feature optimization when constructing the Mahalanobis space based on each of the seven fault modes.

#### 3.2.3. Fault Mode Recognition

According to the selected important features, each Mahalanobis space is reconstructed. The MD from each test sample to each Mahalanobis space is calculated. The fault mode corresponding to the minimum MD is the mode category predicted by the test sample. Sixty test samples were obtained for each category of fault mode. Under the working condition of D_70_BAN, the confusion matrix of the mode recognition results of the test samples is shown in Figure 6. When the imbalance ratio is 9, the accuracy of intelligent fault diagnosis of industrial robots based on MMTS under various working conditions is shown in Table 5. The average diagnostic accuracy is 99.44%.

### 3.3. Comparative Analysis

In this section, the BP, SVM, KNN, C4.5 and RF algorithms, based on the dimension reduction in the principal component analysis, were used for the comparison to evaluate the superiority of MMTS in dealing with data imbalance problem. Conventional diagnostic methods generally use the accuracy as the model evaluation index, but when the samples show an imbalanced distribution, the diagnostic accuracy of the minority categories has little effect on the overall diagnostic accuracy, while the diagnostic accuracy of the majority categories plays a dominant role. It is insufficient to use the diagnostic accuracy alone as an evaluation index for the imbalanced samples. The G-mean takes into account the precision of the minority categories and the precision of the majority categories, and the G-mean is large, but only when both values are large. Therefore, the G-mean can reasonably evaluate the overall classification performance of the imbalanced data. The F-measure incorporates the recall and the precision. The F-measure of the minority categories is large only when both the recall and the precision of the minority categories are large. Therefore, the F-measure can correctly reflect the classification performance of the minority categories. In summary, it is more convincing to discuss the performance of the diagnostic methods by using the accuracy, G-mean and F-measure together as the evaluation metrics in this paper [16]. The comparison results are shown in Figure 7, Figure 8 and Figure 9. For the problem of imbalanced data, the MMTS algorithm proposed in this article has a higher diagnostic accuracy, G-mean and F-measure than the other algorithms under the six working conditions. The average evaluation indexes of the six working conditions are shown in Table 6. The accuracy, F-measure and G-mean of the proposed approach improved by an average of 20.74%, 12.85% and 21.68%, compared with the other five traditional approaches. In summary, the MMTS has good diagnostic performance. It is suitable for solving the fault diagnosis of industrial robots in the actual industrial environment.

To prove the MMTS’ suitability for solving imbalanced data problems, intelligent fault diagnosis schemes with different imbalance ratios were designed in this section, as shown in Table 7. Figure 10, Figure 11 and Figure 12 list the diagnostic accuracy, G-mean and F-measure results of the fault diagnosis algorithms with different imbalance ratios. It can be seen from the figure that the diagnostic accuracy, G-mean, and F-measure of BP, SVM, KNN, C4.5 and RF decrease with the increase in the imbalance ratio. The MMTS algorithm has little difference in diagnostic accuracy, G-mean and F-measure with the increase in the imbalance ratio. It indicates that the imbalance ratio of the data has little influence on the MMTS algorithm. Therefore, for the fault diagnosis research of imbalanced data, the proposed MMTS algorithm in this article has better applicability.

## 4. Conclusions

Addressing the imbalanced data problem faced in the field of industrial robot fault diagnosis, this article proposes an intelligent fault diagnosis approach for industrial robots based on MMTS. With this method, the key features are selected through the OAs and S/N ratios. The Mahalanobis space is reconstructed based on the selected key features. Then, the MD is used as the measurement scale for fault recognition. In order to characterize the effectiveness of the fault diagnosis algorithm comprehensively and reasonably, the diagnostic accuracy, G-mean and F-measure are used as the evaluation indexes of the experiment. The experimental results show that the industrial robot intelligent fault diagnosis approach, based on the MMTS, has obvious advantages compared with the BP, SVM, KNN, C4.5 and RF algorithms under the six working conditions. With the increase in the imbalance ratio, the industrial robot intelligent fault diagnosis approach, based on MMTS, has better diagnosis results and stability. In summary, the fault diagnosis approach proposed in this paper has been validated on industrial equipment. It can bring equally promising diagnostic results in the diagnostic studies of medical diseases, fingerprint recognition and product defects. In future research work, we will continue to study the MMTS in health assessment, life prediction and other related work scenarios to prove the capability of the MMTS. In addition, integrating the MMTS with deep learning methods to improve the performance of the MMTS is also a research direction to be considered in the future.

## Figures and Tables

**Figure 1 entropy-24-00871-f001:**
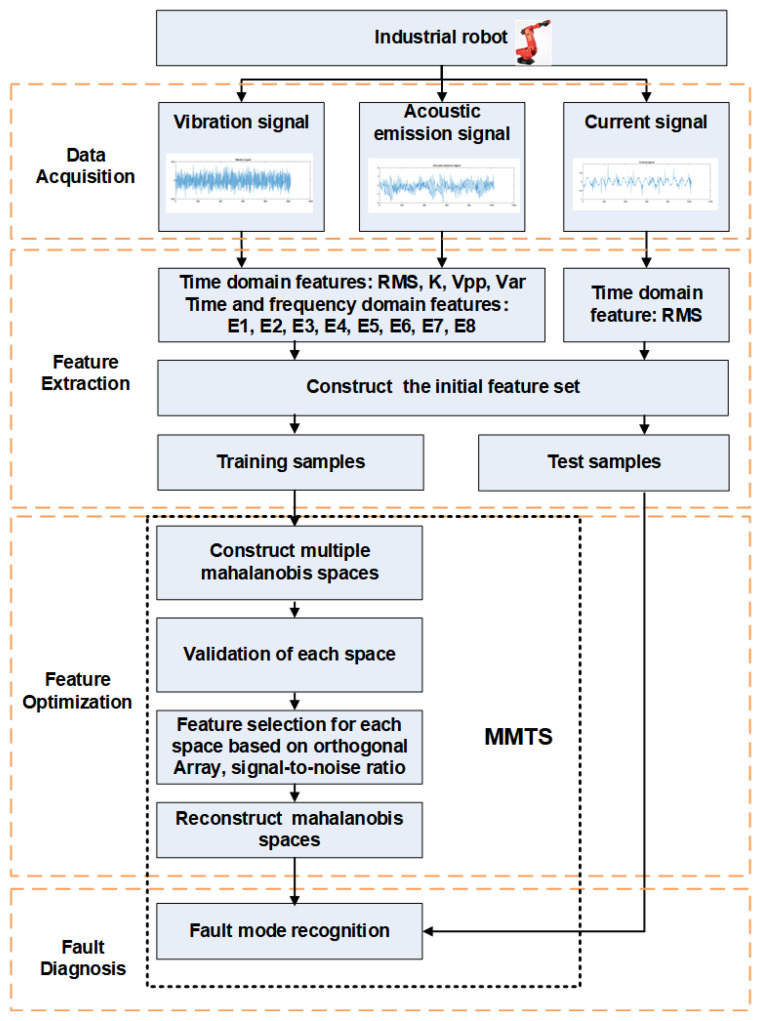
Fault diagnosis framework for the industrial robots based on MMTS.

**Figure 2 entropy-24-00871-f002:**
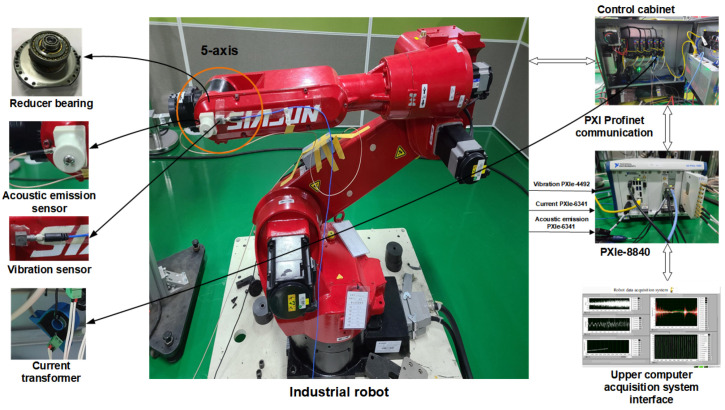
Industrial robot platform.

**Figure 3 entropy-24-00871-f003:**
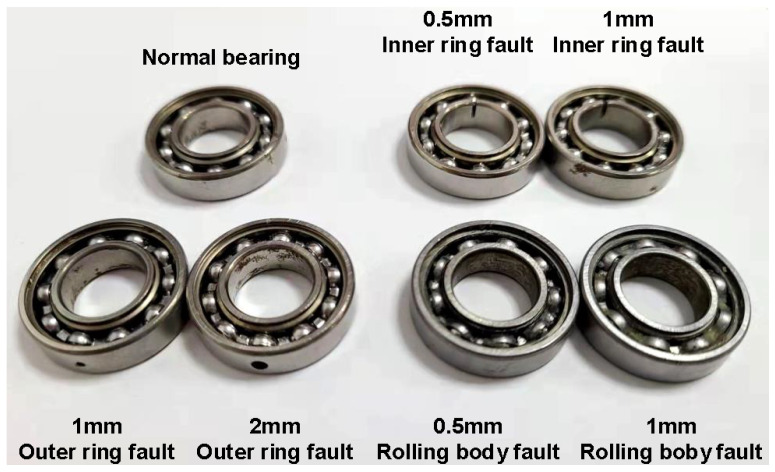
Bearing with 7 fault modes.

**Figure 4 entropy-24-00871-f004:**
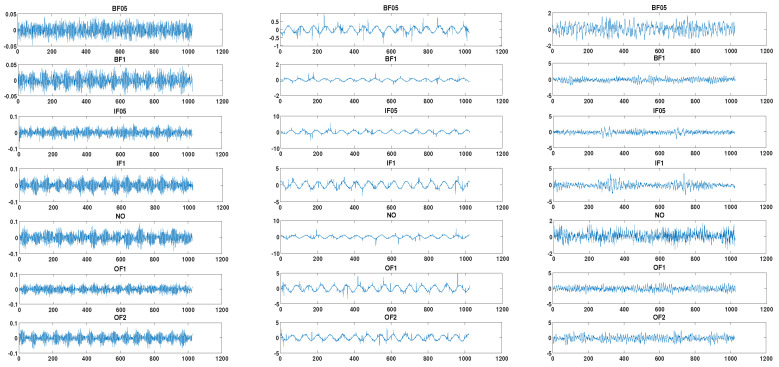
Acoustic emission signal, current signal and vibration signal for 7 modes.

**Figure 5 entropy-24-00871-f005:**
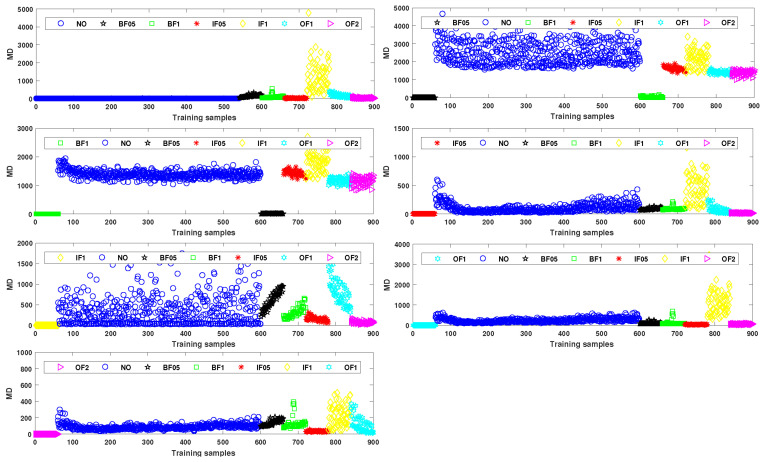
MD with NO, BF05, BF1, IF05, IF1, OF1 and OF2 as the benchmark space, respectively.

**Figure 6 entropy-24-00871-f006:**
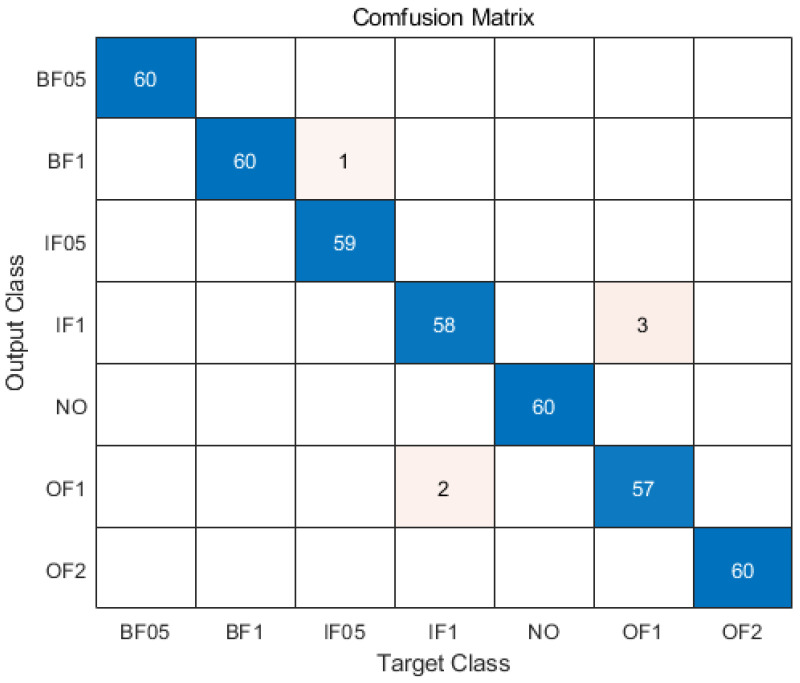
Mode recognition results of test sample.

**Figure 7 entropy-24-00871-f007:**
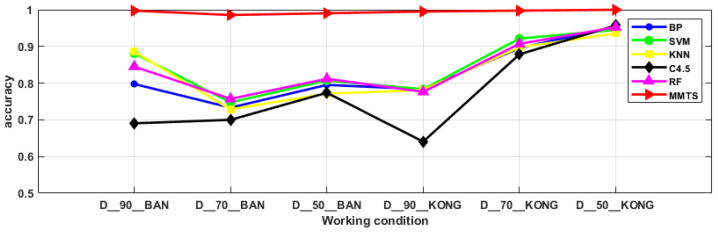
Diagnostic accuracy of six algorithms under six working conditions.

**Figure 8 entropy-24-00871-f008:**
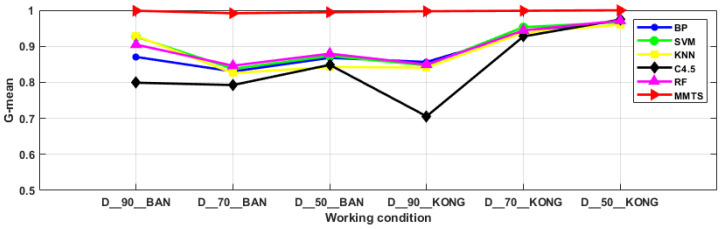
G-mean of six algorithms for six working conditions.

**Figure 9 entropy-24-00871-f009:**
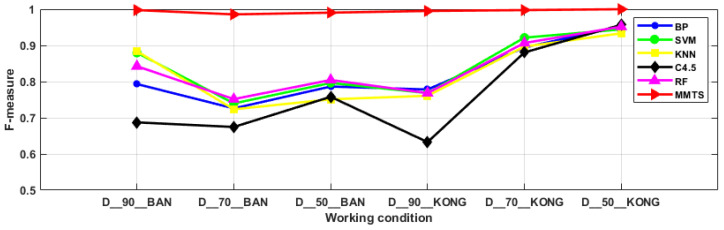
F-measure of six algorithms for six working conditions.

**Figure 10 entropy-24-00871-f010:**
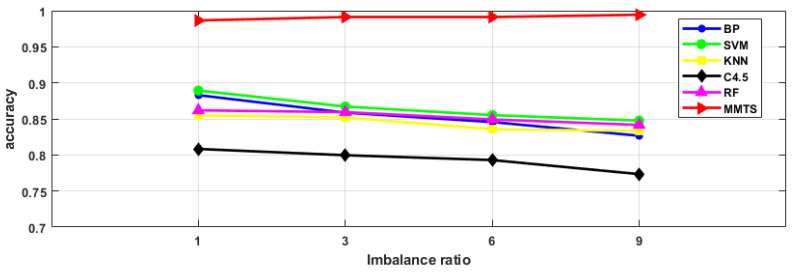
Diagnostic accuracy of the six algorithms under different imbalance ratios.

**Figure 11 entropy-24-00871-f011:**
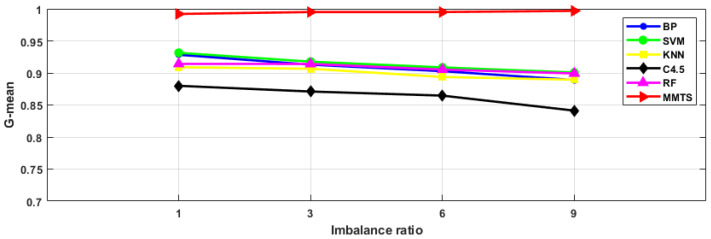
G-mean of six algorithms with different imbalance ratios.

**Figure 12 entropy-24-00871-f012:**
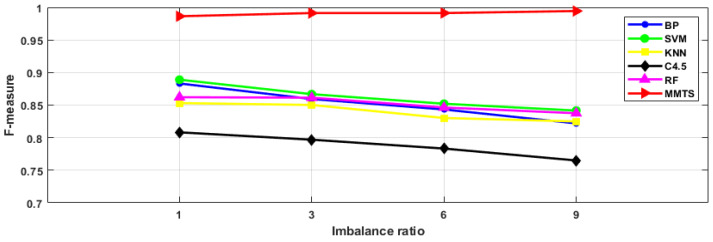
F-measure of six algorithms with different imbalance ratios.

**Table 1 entropy-24-00871-t001:** Description of fault modes.

Fault Mode Indication Symbol	Fault Mode Description
NO	Normal
BF05	0.5 mm pitting fault of rolling body
BF1	1 mm pitting fault of rolling body
IF05	0.5 mm crack fault of inner ring
IF1	1 mm crack fault of inner ring
OF1	1 mm pitting fault of outer ring
OF2	2 mm pitting fault of outer ring

**Table 2 entropy-24-00871-t002:** Introduction of working conditions.

Work Condition Symbol	Rotational Speed	Load
D_90_BAN	90%	Half load
D_90_KONG	90%	No-load
D_70_BAN	70%	Half load
D_70_KONG	70%	No-load
D_50_BAN	50%	Half load
D_50_KONG	50%	No-load

**Table 3 entropy-24-00871-t003:** Optimization results of OAs.

Feature Parameters													
	F1	F2	F3	F4	F5	F6	F7	F8	F9	F10	F11	F12	F13
$SN_{i}^{+}{}^{\left(t\right)}$	14.56	13.532	13.364	14.629	14.181	13.314	13.355	13.197	13.114	13.307	13.596	13.127	14.151
$SN_{i}^{-}{}^{\left(t\right)}$	12.325	13.353	13.521	12.256	12.705	13.571	13.53	13.688	13.772	13.579	13.289	13.758	12.734
$\Delta_{i}^{\left(t\right)}$	2.235	0.179	−0.157	2.373	1.476	−0.257	−0.175	−0.491	−0.658	−0.272	0.307	−0.631	1.417
	F14	F15	F16	F17	F18	F19	F20	F21	F22	F23	F24	F25	
$SN_{i}^{+}{}^{\left(t\right)}$	14.267	14.529	13.9	14.209	13.441	13.327	13.447	13.462	13.304	13.228	13.479	13.261	
$SN_{i}^{-}{}^{\left(t\right)}$	12.618	12.356	12.985	12.676	13.444	13.558	13.439	13.423	13.581	13.657	13.406	13.624	
$\Delta_{i}^{\left(t\right)}$	1.649	2.173	0.915	1.533	−0.003	−0.231	0.008	0.039	−0.277	−0.429	0.073	−0.363	

**Table 4 entropy-24-00871-t004:** Optimized feature subsets.

Fault Modes for the Construction of Mahalanobis Space	Optimized Feature Subset	Optimized Number of Features
NO	[1, 2, 4, 5, 11, 13, 14, 15, 16, 17, 20, 21, 24]	13
BF05	[1, 2, 3, 4, 6, 8, 9, 10, 13, 17, 18, 20, 22, 23, 24, 25]	16
BF1	[1, 2, 3, 4, 6, 8, 9, 10, 13, 17, 22, 24]	12
IF05	[1, 2, 4, 5, 7, 8, 9, 12, 13, 14, 16, 17, 18, 19, 20, 21, 22, 23, 24, 25]	20
IF1	[1, 2, 4, 5, 6, 10, 13, 16, 18, 19, 20, 21, 22, 23, 24, 25]	16
OF1	[1, 2, 4, 5, 9, 12, 13, 14, 15, 16, 17, 18, 19, 21, 22]	15
OF2	[1, 4, 5, 6, 8, 9, 13, 14, 15, 16, 17, 18, 19, 21, 22, 23, 24, 25]	18

**Table 5 entropy-24-00871-t005:** Diagnostic accuracy of the six working conditions based on MMTS algorithm.

Work Condition	Accuracy
D_90_BAN	99.76%
D_90_KONG	99.52%
D_70_BAN	98.57%
D_70_KONG	99.76%
D_50_BAN	99.05%
D_50_KONG	100%

**Table 6 entropy-24-00871-t006:** Diagnostic results of the six algorithms.

	Average Accuracy	Average G-Mean	Average F-Measure
PCA-BP	0.8266	0.889	0.8219
PCA-SVM	0.8476	0.9007	0.8416
KNN	0.8333	0.8894	0.8249
C4.5	0.7734	0.8411	0.7650
RF	0.8416	0.8992	0.8375
MMTS	0.9944	0.9968	0.9944

**Table 7 entropy-24-00871-t007:** Basic description of the dataset.

Dataset	Robot Bearing Fault Data			
Category	7			
Number of features	25			
Number of normal samples (NO)	60	180	360	540
Number of fault samples (BF05, BF1, IF05, IF1, OF1, OF2)	60, 60, 60, 60, 60, 60			
Imbalance ratio	1	3	6	9

## Data Availability

Not applicable.

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
