# Peer review of "Intelligent Fault Diagnosis of Industrial Robot Based on Multiclass Mahalanobis-Taguchi System for Imbalanced Data"

_entropy, 2022, doi:10.3390/e24070871_

Round 1
Reviewer 1 Report
The authors present a multiclass Mahalanobis-Taguchi System to classify fault in industrial manipulators as a method to cope with unbalanced data. They discussed SoA alternative methods and proposed a solution with the feature to overcome overfitting problems and computational costs. The proposed solution is discussed and tested in experimental tests with an industrial robot manipulator where the task was to classify bearing faults. Although the topic of fault diagnosis in the presence of unbalanced data has been widely addressed in the literature, the proposed method represents an alternative and interesting solution with high potential.
The authors could provide in the Conclusion section some hints about possible other applications or improvements to stimulate the research community.
Reviewer 2 Report
Fault diagnostic when only imbalanced data are provided is serious problem. The manuscript is relevant and well organized. However, some questions should be answered or manuscript should be updated:
- The statement: „The traditional diagnosis approaches of industrial robots are more biased to the majority categories, which makes the diagnosis accuracy of the minority categories decrease.“ Should be more detailed. Please, write 2-3 ML methods which are more biased. Please, add reference at the end of respective statement in Introduction (lines 47-50).
- For model accuracy you use G-mean and F-measure. Why you are using only these metrics?
- Abstract should include more numbers which could prove your brief findings stated in the Abstract.
- Why Current signal was aggregated with only RMS while Vibration and Acoustic emission signals were aggregated with more parameters?
- Table 1 is unnecessary.
- Table 2 contains with two “Normal” modes.
- “Normal” mode is not fault , so you have 6 fault modes and 1 normal. I suggest to rethink phase of “7 fault modes”.
- Table 4 is unnecessary.
- Section 3.2.3 must be expanded. Here I am expecting to see overall accuracy (regardless working conditions) and summary table of accuracy metrics for each working condition.
- There are no listed fault modes in Figure 6.
- Random forest is one of the most accurate model in similar experiments. Why haven’t you compared it? Why you didn’t compare any kind of ANN?
- At the beginning of manuscript you are telling that “<…> conventional approaches are more biased <…>” and it is mostly true. Regarding this deficiency you have proposed MMTS. However, Table 8, Figures 10-12 show that MMTS is better than traditional ML then imbalance ratio is 1. That means, that your approach is better whatever the imbalance ratio is. So, the problem is not an imbalance but model application, data processing or other reasons. Please, investigate and show why MMTS is better than other MLs regardless of imbalance ration.
Round 2
Reviewer 2 Report
Accept in present form.